# Longterm Increased S100B Enhances Hippocampal Progenitor Cell Proliferation in a Transgenic Mouse Model

**DOI:** 10.3390/ijms23179600

**Published:** 2022-08-24

**Authors:** Leticia Rodrigues, Krista Minéia Wartchow, Michael Buchfelder, Diogo Onofre Souza, Carlos-Alberto Gonçalves, Andrea Kleindienst

**Affiliations:** 1Department of Biochemistry, Federal University of Rio Grande do Sul, Porto Alegre 90035003, RS, Brazil; 2Department of Neurosurgery, Friedrich-Alexander University, 91054 Erlangen, Germany

**Keywords:** S100B protein, transgenic animals, brain injury, regenerative medicine, hippocampus, adult neurogenesis, therapeutic application

## Abstract

(1) The neurotrophic protein S100B is a marker of brain injury and has been associated with neuroregeneration. In S100Btg mice rendering 12 copies of the murine S100B gene we evaluated whether S100B may serve as a treatment option. (2) In juvenile, adult, and one-year-old S100Btg mice (female and male; *n* = 8 per group), progenitor cell proliferation was quantified in the subgranular zone (SGZ) and the granular cell layer (GCL) of the dentate gyrus with the proliferative marker Ki67 and BrdU (50 mg/kg). Concomitant signaling was quantified utilizing glial fibrillary acidic protein (GFAP), apolipoprotein E (ApoE), brain-derived neurotrophic factor (BDNF), and the receptor for advanced glycation end products (RAGE) immunohistochemistry. (3) Progenitor cell proliferation in the SGZ and migration to the GCL was enhanced. Hippocampal GFAP was reduced in one-year-old S100Btg mice. ApoE in the hippocampus and frontal cortex of male and BDNF in the frontal cortex of female S100Btg mice was reduced. RAGE was not affected. (4) Enhanced hippocampal neurogenesis in S100Btg mice was not accompanied by reactive astrogliosis. Sex- and brain region-specific variations of ApoE and BDNF require further elucidations. Our data reinforce the importance of this S100Btg model in evaluating the role of S100B in neuroregenerative medicine.

## 1. Introduction

Following brain injury, up-regulation of mitogenic and neurotrophic factors represents an endogenous repair mechanism. One of these factors is S100B, a calcium-binding protein secreted by astrocytes and exerting intra- and extracellular regulatory activities [1,2]. Increased extracellular S100B levels are observed during brain development [3], and in a variety of brain pathologies including traumatic brain injury, ischemic insult, and neurodegenerative diseases (see [4] for a review). At nanomolar concentrations, S100B conveys neuroprotective and neurotrophic properties, while at higher concentrations it has been associated with deleterious effects, and may form the basis of neurodegenerative diseases such as Alzheimer’s Disease (AD) [1,4,5]. S100B effects are mediated through the receptor for advanced glycation end products (RAGE) [1]. Persistent RAGE activation causes neuronal death as a result of increased production of reactive oxygen species [6].

Transgenic mouse models overexpressing S100B have initially been developed to study its degenerative effects [7,8], and supported the idea of S100B as a marker of brain damage and neurodegenerative diseases [9,10]. On the other hand, the S100B-induced promotion of hippocampal progenitor cell proliferation following experimental brain injury has received less attention [11,12]. Recently, we reported increased S100B levels in the brain tissue, cerebrospinal fluid (CSF), and serum of S100B transgenic animals [5], and have pursued further research to understand the biochemical effects of S100B overexpression. We refer to the term “long-term increased S100B” levels throughout the text with regard to conceivable (patho) physiological effects, keeping in mind that the transgenic animals overexpress the protein and that the resulting S100B levels range in the nM order, far below toxic μM levels used in cell culture [1]. In the present study, we continue earlier acute studies demonstrating nM S100B levels to enhance hippocampal neurogenesis and cognitive function following experimental brain injury and, even more importantly, in naïve animals [11,13,14,15].

Cognitive impairment is one feature of even mild traumatic brain injury since the hippocampus—the brain region most critical for learning and memory—is especially susceptible to injury [16,17,18]. Hippocampal progenitor cells residing in the subgranular zone (SGZ) of the dentate gyrus in adult mammals provide an intrinsic regenerative potential in the brain [19,20]. The increased proliferation and neuron formation within the hippocampus following brain injury [21] provides an innate repair mechanism of the brain. In addition to growth factors such as brain-derived neurotrophic factor (BDNF) up-regulating neurogenesis [22], S100B contributes to hippocampal network repair after brain injury as well as memory consolidation during development [23].

Another astroglial protein involved in neurogenesis and neurite extension is apolipoprotein E (ApoE) [24]. In the brain, ApoE in the postnatal period is produced predominantly by astrocytes delivering cholesterol to neurons during membrane synthesis, neurite outgrowth, and repair [25]. The isoforms ApoE2 and ApoE3 positively modulate the expression of BDNF [26].

Glial fibrillary acidic protein (GFAP) is a cytoskeletal intermediate filament component commonly interpreted as a marker of mature astrocytes or astroglial reactivity. In the postnatal development of rodents, an age-dependent increase in mRNA and protein is evident [27,28]. In transgenic animals, GFAP participates in postnatal neurogenesis. In the mouse forebrain, postnatal neurons have been demonstrated to derive from distinctive GFAP-positive precursor cells [29,30].

Since we demonstrated that the acute intrathecal and intraperitoneal treatment with S100B in wild-type rodents promoted hippocampal neurogenesis and cognitive function [11,14,15], we utilized the S100Btg model to elucidate the effect of long-term increased S100B levels. Specifically, we quantified adult progenitor cell proliferation in the germinative area of the dentate gyrus (SGZ), as well as in the area of the destination of their migration (granular cell layer, GCL). in juvenile, adult, and one-year-old mice, rendering 12 copies of the murine S100B gene [7]. To characterize involved cellular events and signaling mechanisms, we quantified the levels of other proteins associated with neurogenesis, BDNF, and ApoE, the astroglial marker GFAP, and the main S100B receptor RAGE.

## 2. Results

To evaluate the effects of long-term increased S100B levels on hippocampal neurogenesis in S100Btg, we quantified the progenitor cell proliferation utilizing the proliferative marker BrdU and Ki67 in juvenile (28 days), adult (3 months), and senile (one year) wt and S100Btg mice. While the most sophisticated method estimating adult neurogenesis is stereological quantification following intraperitoneal injection of the mitotic marker BrdU [31], we realized in our earlier studies the daunting requirements in terms of resources [11]. Furthermore, BrdU injections constitute an experimental burden to the animals increasingly addressed in the European Union (compare Directive 2010/63/EU on the protection of animals used for scientific purposes). Hence, we compared the standard method following BrdU injections and Ki67 immunohistochemistry demonstrated to correlate significantly [32]. Involved cellular events were assessed quantifying ApoE and BDNF levels in the blood serum, CSF, adipose tissue, and different brain regions by ELISA as well as the hippocampal content of GFAP and RAGE by Western blotting.

### 2.1. Effect of Long-Term Increased S100B Levels on Hippocampal Progenitor Cell Proliferation and Migration

The comparison of the proliferative marker BrdU and Ki67 for quantification of progenitor cell proliferation revealed that both markers provided highly significant congruent results (Figure 1A; BrdU mean 2.33 ± 2.99 immunoreactive cells; F_5,210_ = 33.91, *p* < 0.001; Ki67 mean 1.53 ± 2.90 immunoreactive cells; F_5,206_ = 73.49, *p* < 0.001; r = 0.608, *p* < 0.001). Hence, for clarity of presentation we continue to refer to the Ki67 data (Figure 1B).

Utilizing Ki67-immunoreactivity to estimate the total number of proliferating cells within the germinative area of the dentate gyrus, the SGZ, we found in S100Btg animals a significant mitogenic response to S100B exposure (F_5,548_ = 9.87, *p* = 0.002) as compared to wt animals. The post-hoc analysis demonstrated that S100B significantly increased the proliferative response in the SGZ (juvenile: S100Btg 6.67 ± 0.36; wt 4.96 ± 0.32, *p* < 0.001; adult: S100Btg 0.78 ± 0.14; wt 0.04 ± 0.02, *p* = 0.027; old: S100Btg 0.24 ± 0.07; wt 0). To determine the fate of the proliferating cell population generated in the germinative area of the dentate gyrus, the SGZ, we quantified the number of proliferating cells in the area of the destination of their migration, the GCL. Long-term S100B exposure promoted the migration to the GCL in older S100Btg mice (juvenile: S100Btg 0.05 ± 0.36; wt 0.05 ± 0.03; adult: S100Btg 0.05 ± 0.03; wt 0.02 ± 0.01; old: S100Btg 0.05 ± 0.02; wt 0; Figure 1B).

### 2.2. Effect of Long-Term Increased S100B Levels on Central and Peripheral ApoE Content

ApoE brain content was quantified in the hippocampus, frontal cortex, and hypothalamus in one-year-old wt and S100Btg mice. In female S100Btg mice, the ApoE was not different in any region (Figure 2A, *p* = 0.843; Figure 2C, *p* = 0.793; Figure 2E, *p* = 0.406). In male S100Btg mice, the ApoE was significantly reduced in the hippocampus and frontal cortex, but not in the hypothalamus as compared to wt mice (Figure 2B, *p* = 0.025; Figure 2D, *p* = 0.003; Figure 2F, *p* = 0.281). The ApoE in serum, CSF, and adipose tissue did not differ between S100Btg and wt mice, either in female (Figure 3A, *p* = 0.821; Figure 3C, *p* = 0.494; Figure 3E, *p* = 0.531) or in male mice (Figure 3B, *p* = 0.625; Figure 3D, *p* = 0.351; Figure 3F, *p* = 0.793).

### 2.3. Effect of Long-Term Increased S100B Levels on Central BDNF Content

BDNF levels were quantified in the hippocampus and frontal cortex in one-year-old wt and S100Btg mice. BDNF was significantly reduced in the frontal cortex but did not differ in the hippocampus of female S100Btg mice, and in any region of male S100Btg mice (Figure 4A, *p* = 0.988; Figure 4B, *p* = 0.657; Figure 4C, *p* = 0.036; Figure 4D, *p* = 0.402).

### 2.4. Effect of Long-Term Increased S100B Levels on Hippocampal RAGE and GFAP Content

GFAP and RAGE content were quantified in the hippocampus in one-year-old wt and S100Btg mice. RAGE content did not differ either in female or in male mice (Figure 5A, *p* = 0.944; Figure 5B, *p* = 0.892). The glial marker GFAP was significantly reduced in one-year-old female and male S100Btg mice (Figure 5C, *p* = 0.026; Figure 5D, *p* = 0.024).

## 3. Discussion

Evidence documents adult neurogenesis and suggests the exogenous application of neurotrophic factors to promote innate repair mechanisms following brain injury. Several lines of evidence indicate that the neurotrophic protein S100B participates in neuroplasticity [33], neurogenesis [6], and other cellular events following brain injury. However, the specific role of S100B is ambiguous, may vary over time, and even contribute to AD [2,34]. Continuing earlier studies demonstrating enhanced hippocampal neurogenesis and improved cognitive function resulting from a short-term S100B treatment at nanomolar concentrations, we aimed to evaluate the effect of long-term increased S100B concentrations. In our S100Btg model rendering 12 copies of the murine S100B gene, we demonstrated the effective S100B concentration to be in the nanomolar range [5].

There are two new findings in this study. First, long-term increased nanomolar S100B levels promote progenitor cell proliferation in the germinative area of the hippocampus (SGZ) in juvenile, adult, and one-year-old S100Btg mice, as well as migration to the GCL in older S100Btg mice. Second, long-term increased nanomolar S100B levels did not promote astrogliosis but rather correlated significantly with decreased hippocampal GFAP content within the investigation period. The sex-dependent reduction of ApoE in the hippocampus and frontal cortex of male S100Btg mice, and of BDNF in the frontal cortex of female S100Btg, require further investigation.

### 3.1. Long-Term Increased S100B Levels Promote Hippocampal Neurogenesis

Previously, we found the acute therapeutic application of S100B at nanomolar concentrations to promote hippocampal progenitor cell proliferation, differentiation, and migration [11], and to improve the functional hippocampal-dependent outcome after experimental brain injury in rats [15]. Furthermore, we were able to demonstrate that the S100B-induced hippocampal neurogenesis was complemented by an enhanced synaptogenesis in rodents, both following injury and in uninjured controls [13,14]. In the present study, we validated the effect of long-term increased nanomolar S100B concentrations in S100Btg mice—as demonstrated in an earlier study [5]—which is to promote hippocampal progenitor cell proliferation in the germinative area of the dentate gyrus, the SGZ, as well as migration to the GCL in older mice.

To assess the proliferative response of brain progenitor cells, in all of our previous studies animals were injected intraperitoneally with the mitotic marker BrdU. Although we cannot exclude that the BrdU-incorporation occurs in part as a result of DNA repair, it has been well documented by others that BrdU-labeling is specific for progenitor cell division and provides an effective tool for measuring neurogenesis [21,35]. However, BrdU injections constitute an experimental burden to the animals. Since the resulting stress may counteract hippocampal neurogenesis [36], we evaluated whether the proliferation marker Ki67 [37] may represent a valuable alternative without requiring repetitive injections in the living animal. Following the literature, the number of BrdU- and Ki67-labeled proliferating cells in the dentate gyrus correlates significantly [32]. However, in contrast to the results of Kee et al., in our study, the number of Ki67 immunoreactive cells represented only two-thirds of the BrdU immunoreactive cells. We attribute our findings to a weaker antibody staining (compare Figure 1A).

In the current study, we examined whether the long-term increased S100B nanomolar levels promote progenitor cell proliferation in the SGZ of the dentate gyrus and proliferation to the GCL. In accordance with the literature, we documented a significant neurogenesis in juvenile wt mice which decreased with age [19,38]. We demonstrated S100B to increase proliferation in juvenile S100Btg mice by 50%, and in adult ones by 200%. While in one-year-old wt animals, no proliferating cells could be detected, in S100B animals 0.24 ± 0.07 immunoreactive cells were labelled in the SGZ. Furthermore, the migration of proliferating progenitor cells was promoted with increasing age. These findings are in line with the literature documenting neurotrophic properties of S100B at nanomolar concentrations [6,39,40].

### 3.2. Long-Term Increased S100B Levels Attenuate Hippocampal and Frontal ApoE Content in Male S100Btg Mice

Brain ApoE is primarily secreted by astrocytes in adult animals, and interacts with neurogenesis and neural plasticity through the delivery of cholesterol to the membrane [24]. Recent studies have confirmed the importance of astroglial ApoE for hippocampal neurogenesis and cognitive recovery following injury [41]. We quantified the ApoE in the hippocampus, frontal cortex, and hypothalamus of one-year-old S100Btg mice. In male S100Btg mice, the ApoE was significantly reduced in the hippocampus and frontal cortex, but not in the hypothalamus as compared to wt controls. No changes were observed in female mice. These findings may be attributed to the regulation of ApoE expression by sex steroids [42], and the sex-dependent ApoE expression—particularly of isoform ApoE4—in AD [43].

It is worth mentioning that the S100B expression in S100Btg mice is increased in all these brain regions, regardless of sex [5]. Hence, the promoted hippocampal progenitor cell proliferation, independent of sex, may be related rather to the increased S100B levels. Since the hippocampal ApoE content but not the spatially correlating progenitor cell proliferation is sex-dependent, we doubt a respective interaction in S100Btg mice. However, these findings do not rule out the possibility that ApoE may actively contribute, in a sex-dependent manner, to hippocampal recovery following injury, and has to be taken into account from a translational perspective. Indeed, an effect on neurogenesis was observed in knock-in mice for human ApoE isoforms, dependent on the isoform, age, and sex of the animal [44].

### 3.3. Long-Term Increased S100B Levels Are Not Associated with Increased Hippocampal BDNF Content

Although BDNF levels in brain tissue may fluctuate in the postnatal period dependent on the brain region and species, in general, levels decrease with age. Furthermore, BDNF expression is affected by sex steroids [45], and critically involved in the regulation of age-related processes in the hippocampus, including neurogenesis and neuronal plasticity [30]. Since ApoE may positively modulate BDNF expression [24], it would be interesting whether the decreased hippocampal ApoE levels in male S100Btg mice found in our study were accompanied by altered BDNF. We quantified BDNF in the hippocampus and frontal cortex of one-year-old S100Btg mice. In female S100Btg mice, BDNF was significantly reduced in the frontal cortex compared to wt controls.

In S100B knockout animals, BDNF has been demonstrated to be increased in the area of adult neurogenesis, the hippocampus, suggesting a cross-talk of BDNF and S100B in their proliferative properties [46]. However, in the present study, we could not verify any change of hippocampal BDNF levels in S100Btg mice. Since increased S100B levels in S100Btg mice have been demonstrated to enhance the susceptibility to environmental stimuli during adolescence, resulting in more variable behavioral and neural phenotypes in adulthood [47], S100B itself rather than BDNF may have more impact on the enhanced hippocampal neurogenesis, and improved functional outcome demonstrated in earlier studies [15].

### 3.4. Long-Term Increased S100B Levels Are Not Accompanied by Enhanced Hippocampal RAGE

RAGE, the main S100B receptor, is addressed as a key element in inflammation and degeneration. In fact, within minutes of brain injury, a significant neuroinflammatory response occurs in vitro and in vivo [48]. Microglia represent the innate immune system of the brain [49], and astroglial S100B has been proposed to propagate microglia activation by engaging RAGE [50]. In line with the existing literature, we demonstrated in a previous study a microglial activation early after brain injury, which was further enhanced by a S100B treatment [13]. Since neuroinflammation has been proposed to be the pathophysiological mechanism underlying chronic neurodegeneration [51,52], we pursued this set of experiments elucidating the effect of long-term increased S100B levels. Both, the neurotoxic effect and the neurotrophic S100B effect are mediated by RAGE [1]. The roles of RAGE in hippocampal neurogenesis and neurite growth are mediated by S100B and other ligands such as High Mobility Group Box 1 (HMGB1) [53]. An upregulation of RAGE may accompany the increased levels of S100B or other RAGE ligands, triggering a proinflammatory circle [54]. Interestingly, we could not confirm an increased RAGE expression in one-year-old S100Btg mice. In neurodegenerative diseases, such as AD, RAGE upregulation has been implicated in the stimulation of neuronal APP expression (e.g., [55]), and APP has been demonstrated to induce microglial activation accompanied by an increased inducible NO synthase expression [56]. We did not find any effect of long-term increased S100B levels on APP expression in one-year-old S100Btg mice (data not shown). Therefore, we considered the activation of the S100B/RAGE pathway in the hippocampus in S100Btg mice without RAGE upregulation, and that this, at least in these animals, promoted neurogenesis.

### 3.5. Long-Term Increased S100B Levels Are Associated with a Reduced Hippocampal GFAP Content

Astrogliosis is closely associated with regions of neuronal cell loss early after brain injury, while later on, reactive astrocytes are also present in the hippocampus [57]. In vitro, the expression of several glial proteins is regulated by neuronal interactions [58]. In contrast, an increased hippocampal GFAP expression may reflect synaptic re-modeling, and is stimulated by an enriched environment [59]. Thus, reactive astrocytes may play a role in supporting regenerative capabilities of the central nervous system [60]. Although gliogenesis within the hippocampus has been reported in transgenic mice expressing increased levels of S100B [7], we did not find that long-term S100B exposure at nanomolar levels increased gliogenesis in one-year-old S100Btg animals, compared to wt controls. Rather, more importantly, the GFAP expression in the hippocampus was significantly reduced in one-year-old female and male S100Btg mice.

To interpret these findings, it is necessary to contemplate the complex modulation of GFAP expression via signal transducer and activator of transcription 3 (STAT3) [61]. In fact, S100B via RAGE is able to modulate STAT3, and therefore, affect GFAP expression. However, the effect of persistent and high S100B levels on the RAGE/STAT3 pathway is largely unknown. In the present study, we did not observe changes in RAGE content. Moreover, we cannot rule out that the effect of S100B on GFAP expression results from an intracellular interaction, as it could affect many targets including other transcription factors, such as p53 [1]. Regardless of the mechanism involved, our data are in agreement with a study in young S100Btg mice demonstrating a reduction in S100B immunoreactive astrocytes, as well as a reduction in the branching of these astrocytes [62]. This astroglial atrophy in S100Btg mice may reflect toxicity of persistently high S100B levels. In contrast, S100B knock-out mice demonstrate an enhanced GFAP immunoreactivity [63]. Whether the reduced GFAP content found in our study results from attenuated protein synthesis or an increased degradation has to be evaluated in future studies, as well as the effect on other intermediate filaments found in glial cells, such as nestin and vimentin.

### 3.6. Limitations

To quantify progenitor cell proliferation, BrdU/Ki67 immunoreactive cells were quantified on brain sections drawn at systematic intervals throughout the hippocampus. These estimates using a uniform random systematic sampling method are much closer to the true value than estimates using a random independent sampling because the possibility of sample clustering is eliminated [64]. Although the quantification was not performed utilizing an unbiased stereological approach, the random systematic sample method is considered to be appropriate in this proof-of-concept study. While Western blotting remains the gold standard means of quantifying tissue protein expression, only immunohistochemistry maintains the anatomical information.

## 4. Materials and Methods

S100Btg cryo-preserved embryos were purchased from Jackson Laboratories, Charles River, Schulzfeld, Germany (C57BL/6J-Tg (S100b) 5.12Rhr/J; stock no 002260) containing 12 copies of the mouse S100B gene [7]. The strain was recovered in the bio-technical laboratory of the University of Erlangen-Nürnberg, and maintained by crossing carriers of Tg (S100B) 5.12Rhr to C57BL/6N mice because the transgene is homozygous lethal. Offspring were characterized by polymerase chain reaction (PCR)-based genotyping for S100B constructs. We used only littermates obtained from this cross-breeding strategy for all analyses. Thus, all mice used in the present study are comparable in terms of genetic background. All animal experiments were approved by the Institutional Animal Care and Use Committee (No. TS-6/14) and were conducted in accordance with the National Research Council’s guide for the care and use of laboratory animals. The mice were kept in a controlled environment (12:12 h light/dark cycle, 22 ± 1 °C with 60% humidity). Pellet food and tap water were available ad libitum.

For mice genotyping, deoxyribonucleic acid (DNA) was isolated from the mice’s tail using HotShot-Tail-Lyse. An amount of 50 µL alkaline lysis solution was added, the samples were boiled at 95 °C for 1 h, and shaken at 1200 rpm. After the samples had cooled, a 50 µL neutralization buffer was added. DNA could directly be used by PCR assay. The DNA for the genes encoding S100B and β-actin was performed using the primer pairs described below, and Power SYBR Green PCR Master Mix (Invitrogen) for electrophoresis analysis. Target: S100B-Sense/anti-sense: oIMR7408-F, 5′-CGAAGTTGAGATTCACAGACG-3′/oIMR7409-R, 5′-ATCATGACTGGGAAGGTTCC-3′-Product size (pb): 500. Target levels were normalized to β-actin levels (for details see [5]).

### 4.1. Effect of Long-Term Increased S100B Levels on Hippocampal Progenitor Cell Proliferation and Migration

To assess the proliferative response of hippocampal progenitor cells, animals were injected intraperitoneally with the mitotic marker 5-Bromo-2′-deoxyuridine-5′-monophosphate (BrdU, 50 mg/kg body weight, Roche, Germany), which was given once daily for 4 days before sacrifice. A stock solution of 50 mg/mL of BrdU dissolved in sterile saline was made fresh daily. For the histological assessment, the brains were blocked for coronal sections and paraffin-embedded for microtome sectioning. Five µm sections were taken using a rotary microtome from the habenular commissure to the dorsal blade of the dentate gyrus, and 5 adjacent sections were collected every 100 µm. The tissue was deparaffinized by submerging the slides in xylene, rehydrated in decreasing concentrations of ethanol (100% twice followed by once each of 95% and 70%), followed by 15 min incubation in 3% hydrogen peroxide. Finally, the tissue was washed in Tris-buffered saline (TBS) for 2 × 5 min. Sections were boiled in 0.01M citrate buffer (pH 6.0; antigen retrieval solution) for 2-min at 900 Watts, followed by 2 × 10 min at 250 Watts, using a microwave oven, and washed twice in TBS for 5 min before blocking with 10% horse serum. After cooling to room temperature, sections were incubated for 5 min in 2N HCl at 37 °C, and washed three times for 10 min using distilled water and PBS respectively. Sections were blocked for 2 h using phosphate-buffered saline (PBS) containing 0.2% Triton and 10% Horse serum, followed by overnight incubation containing rat anti-BrdU (1:100, Abcam ab6326, Germany) at 4 °C. The sections were washed 3 × 10 min and incubated in PBS containing goat anti-rat conjugated with Alexa-Flour 647 (1:200, Abcam ab150167, Berlin, Germany) secondary antibody.

To test whether the burden resulting from BrdU injection in experimental animals could be reduced, sections adjacent to those stained for BrdU were processed with an antibody against the proliferation marker Ki67 (rabbit anti-Ki67 1:50, Abcam ab16667, Germany; secondary antibody donkey anti-rabbit Alexa-Flour 647 1:400, dianova 711-605-152, Germany). Counterstaining was performed with Invitrogen ProLong Gold antifade reagent with 4′,6-diamidino-2-phenylindole (DAPI; Thermo Fisher Scientific, Erlangen, Germany). Sections were visualized using a fluorescent microscope (Axio Observer.Z1 with ApoTome and AxioCam MR R3, Zeiss, Aalen, Germany). Since the number of BrdU and Ki67 immunoreactive cells in all juvenile experimental groups (male and female, wt and S100Btg) were highly significantly correlated (see below), we performed only Ki67 immunohistochemistry in the subsequent age groups (adult and one-year-old animals).

### 4.2. Quantification of Hippocampal Progenitor Cell Proliferation

BrdU and Ki67 immunoreactive cells were quantified on brain sections drawn at uniform systematic intervals throughout the entire hippocampus starting at a random point. These estimates using a uniform random systematic sample method are much closer to the true value than estimates using a random independent sample, because the possibility of sample clustering is eliminated [62]. Although the quantification was not performed utilizing an unbiased stereological approach, the random systematic sample method was considered to be appropriate in this proof-of-concept study. In every section sampled, the contour of the dentate gyrus was delineated, and due to the relatively low numbers of proliferating cells, BrdU- and Ki67-immunoreactive cells in the entire area comprising of the germinative area of the dentate gyrus, the subgranular zone (SGZ), and the destination of their migration, the granular cell layer (GCL), were quantified. A total of 10 to 12 sections per animal were included in the group comparison, and cell counting was performed by two independent investigators, who were blinded to the experimental group. The area of the dentate gyrus was quantified using Cell Imaging software version 1.16 (Olympus Soft imaging solutions GmbH, Münster, Germany).

### 4.3. Enzyme-Linked Immunosorbent Assay for ApoE Content

The measurement of ApoE content was performed using a commercial mouse enzyme-linked immunosorbent assay (ELISA, LSbio LS-F4714, Seattle, WA, USA), combined with colorimetric detection. Tissue and cerebrospinal fluid (CSF) were prepared according to the manufacturer’s instructions. Each well of the supplied microtiter plate was pre-coated with a target-specific capture antibody. Standards or samples are added to the wells and the target antigen binds to the capture antibody. An avidin-horseradish peroxidase (HRP) conjugate is then added, which binds to the biotin. A 3, 3′, 5, 5′-tetramethylbenzidine (TMB) substrate is then added which reacts with the HRP enzyme resulting in color development. Finally, the absorbance was determined at 450 nm, detection Range 15.63–1000 ng/mL. Protein content was measured by a bicinchoninic acid (BCA) protein assay kit (Thermo Fisher Scientific, Waltham, MA, USA), according to the manufacturer’s instructions.

### 4.4. Enzyme-Linked Immunosorbent Assay for BDNF Content

The measurement of BDNF content was performed using a commercial mouse ELISA (MyBiosource MBS355435, San Diega, CA, USA), combined with colorimetric detection. The tissue was prepared according to the manufacturer’s instructions. This kit was based on sandwich enzyme-linked immune-sorbent assay technology. An anti-BDNF polyclonal antibody was pre-coated onto 96-well plates. The biotin conjugated anti-BDNF polyclonal antibodies were used as detection antibodies. The standards, test samples, and biotin-conjugated detection antibodies were subsequently added to the wells. The avidin–biotin–peroxidase complex was added, and unbound conjugates were washed. TMB substrates were used to visualize HRP enzymatic reactions. TMB was catalyzed by HRP to produce a blue color product that changed into yellow after adding an acidic stop solution. The density of yellow is proportional to the BDNF amount of the sample captured in the plate. The optical density (O.D.) absorbance was read at 450 nm in a microplate reader, detection range 31.2–2000 pg/mL. Protein content was measured by a BCA protein assay kit (Thermo Fisher Scientific, USA), according to the manufacturer’s instructions.

### 4.5. Western Blot Analysis for RAGE and GFAP

Hippocampal samples were homogenized in sample buffer (Novex™ Tris-Glycine sodium dodecyl sulfate (SDS) sample buffer-LC2676-and 5% (*w*/*v* β-mercaptoethanol) and separated by SDS-polyacrylamide gel electrophoresis (PAGE) on 12% (*w*/*v*) acrylamide gel, before electro transferring onto nitrocellulose membranes. Membranes were incubated in TRIS-buffered saline (TBS)-Tween (20 mmol/L Tris–HCl, pH 7.5, 137 mmol/L NaCl, 0.05% (*v*/*v*) Tween 20) containing 5% (*w*/*v*) bovine serum albumin (BSA) for 2 h at agitation and at room temperature. Subsequently, the membranes were incubated overnight with the respective primary antibodies anti-RAGE (1:2500 Millipore, Burlington, MA, USA), anti-GFAP (1:5000 Dako, Santa Barbara, CA, USA), and anti- β-actin (dilution 1:10,000 Proteintech, Rosemont, IL, USA), rinsed with TBS-T, and exposed to HRP-linked anti-IgG antibodies for 2 h at room temperature. Chemiluminescent bands were detected using Imagequant LAS4000 GE Healthcare, and densitometric analyses were performed using Image-J software version 1.53 (National Institutes of Health, Bethesda, MD, USA). Results are expressed as percentages of the control.

### 4.6. Statistical Analysis

Data are presented as means ± standard error of mean (SEM). In vivo experiments used eight animals per group. The data were analyzed by Student’s *t*-test. Values of *p* < 0.05 were considered significant. All analyses were performed using the Graphpad Prism software version 8 (Dotmatics, La Jolla, CA, USA).

## 5. Conclusions

In summary, these data reinforce the possibility that nanomolar extracellular S100B levels stimulate hippocampal neurogenesis, and that even long-term exposure to S100B does not result in reactive or adaptive astrogliosis. Sex- and brain region-specific variations of ApoE and BDNF levels require further elucidation. Our data emphasize the importance of this S100B transgenic model in studying the effect of long-term exposure to S100B, to elucidate its specific role in neurodegenerative and/or neuroadaptive processes.

## Figures and Tables

**Figure 1 ijms-23-09600-f001:**
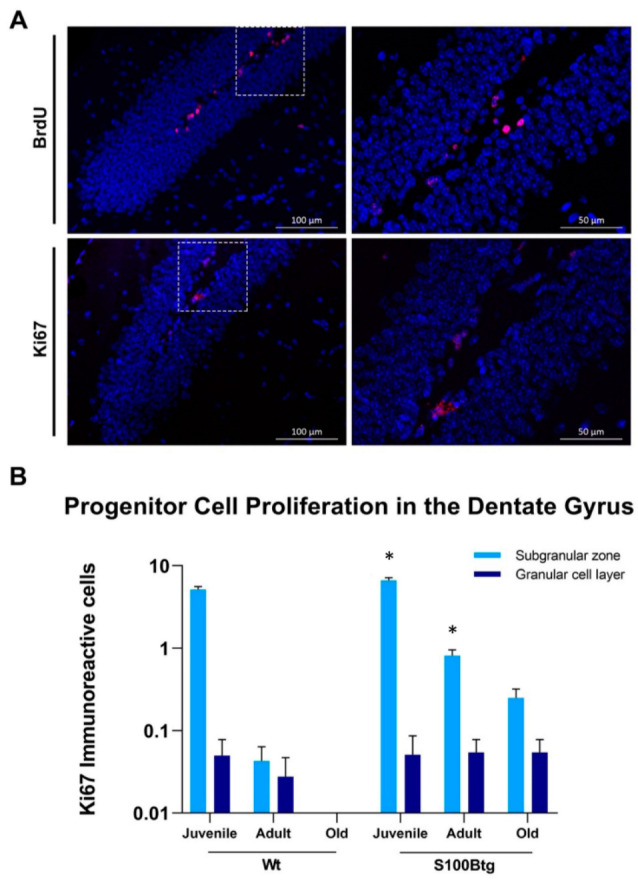
Progenitor cell proliferation in the dentate gyrus in S100Btg and wt mice. (**A**) Comparison of BrdU and Ki67 immunoreactive cells in the dentate gyrus; (**B**) Quantification of progenitor cell proliferation in the dentate gyrus following Ki67 immunostaining. Consider the logarithmic scale. Data are expressed as means ± SE (8 mice/group). * Indicates a statistically significant difference in comparison to wt controls (Student’s *t*-test, assuming *p* < 0.05).

**Figure 2 ijms-23-09600-f002:**
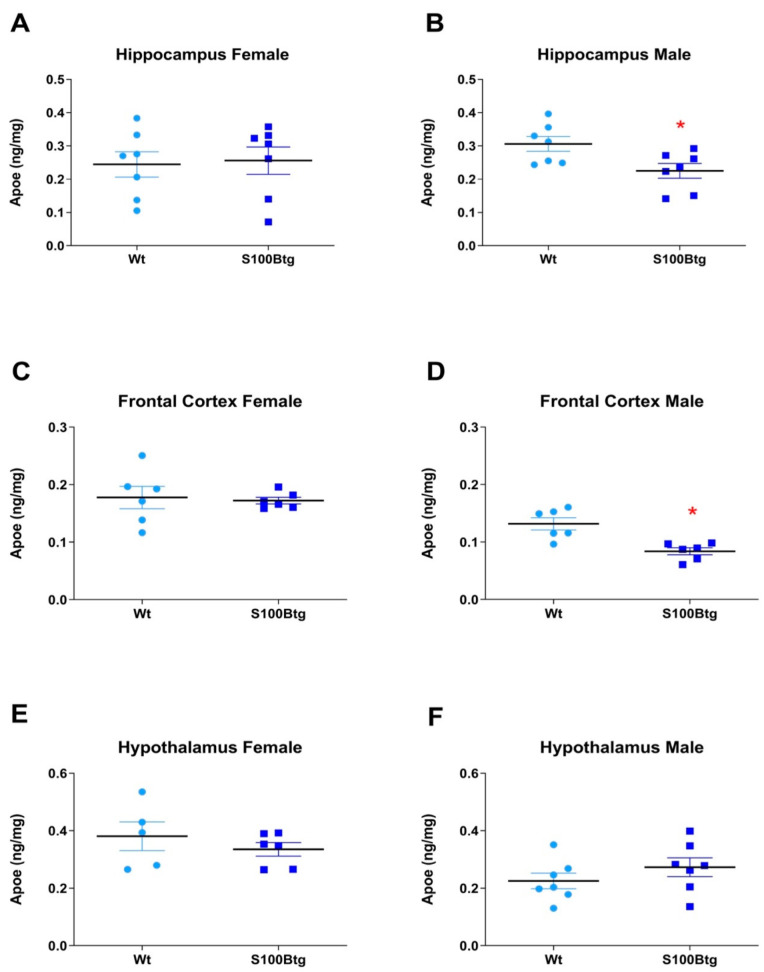
ApoE brain content in one-year-old S100Btg and wt mice. (**A**,**B**) hippocampus, (**C**,**D**) frontal cortex, and (**E**,**F**) hypothalamus; measurements were performed by ELISA. Data are expressed as means ± SE (7 mice/group). * Indicates a statistically significant difference (Student’s *t*-test, assuming *p* < 0.05).

**Figure 3 ijms-23-09600-f003:**
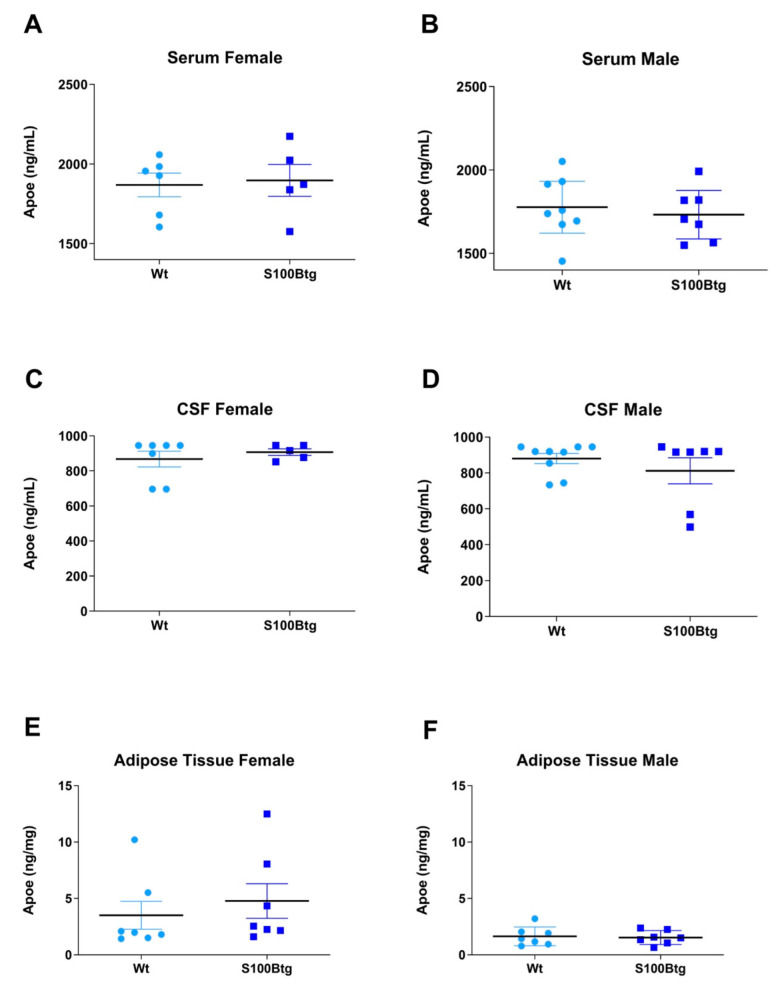
ApoE level in blood serum, cerebrospinal fluid, and adipose tissue in one-year-old S100Btg and wt mice. (**A**,**B**) serum, (**C**,**D**) cerebrospinal fluid, and (**E**,**F**) adipose tissue; measurements were performed by ELISA. Data are expressed as means ± SE (7 mice/group).

**Figure 4 ijms-23-09600-f004:**
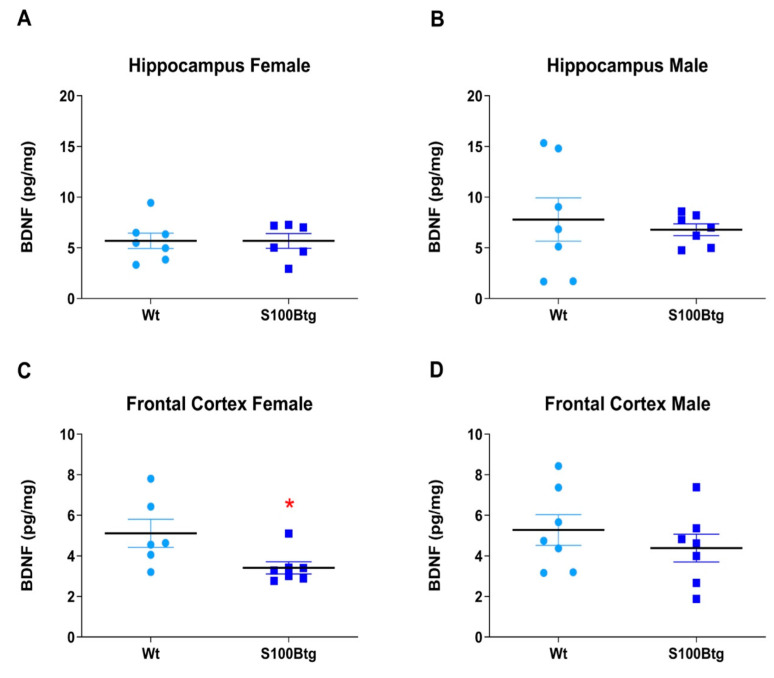
BDNF brain content in one-year-old S100Btg and wt mice. (**A**,**B**) hippocampus, and (**C**,**D**) frontal cortex; measurements were performed by ELISA. Data are expressed as means ± SE (7 mice/group). * Indicates a statistically significant difference (Student’s *t*-test, assuming *p* < 0.05).

**Figure 5 ijms-23-09600-f005:**
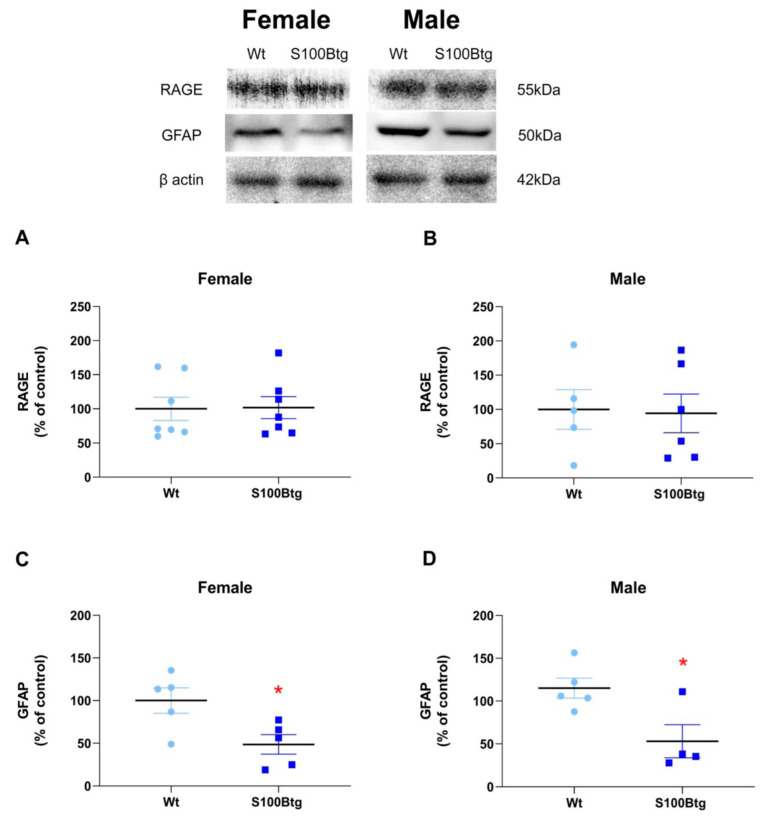
Hippocampal RAGE and GFAP content in one-year-old S100Btg and wt mice. (**A**,**B**) RAGE, and (**C**,**D**) GFAP; analyzed by Western blotting, and protein bands quantified by densitometry. Data are expressed as means ± SE (7 mice/group). * Indicates a statistically significant difference (Student’s *t*-test, assuming *p* < 0.05).

## Data Availability

The data presented in this study are available on request from the corresponding author.

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
