# Peer review of "Longterm Increased S100B Enhances Hippocampal Progenitor Cell Proliferation in a Transgenic Mouse Model"

_ijms, 2022, doi:10.3390/ijms23179600_

Round 1
Reviewer 1 Report
This is a very nice contribution of Dr L Rodrigues on transgenic mice overexpressing S100B protein. Assuming this is a model of long-term exposure to S100B, authors claim that S100tg mice have enhanced hippocampal progenitor cell proliferation, an effect that was not accompanied by reactive astrogliosis.
Experimental design is straightforward and methods used are adequate. Briefly, juvenile, adult and old-aged (1-y-old) Wt and Tg mice had cell proliferation estimated in the subgranular zone and the granular cell layer of the dentate gyrus, as assessed by BrdU and Ki67 markers. GFAP, ApoE, BDNF and RAGE were also studied, with enzymatic or western blot methods. There was an increase in progenitor cell proliferation in the subgranular zone in all age groups studied. Signaling was studied in 1-y-old mice; a decrease of ApoE content was found in the hippocampus and frontal cortex of Tg males, although its levels in serum, CSAF and in the adipose tissue did not differ. BDNF content decreased in the frontal cortex, but not in the hippocampus, of females, and GFAP hippocampus levels were decreased on both male and female Tg mice. Authors conclude that enhanced dentate neurogenesis in S100Tg mice is not accompanied by reactive astrogliosis and that sex dimorphism on the effects measured on ApoE and BDNF require further studies.
Results are original and shed light on S100B actions in the Tg model that might be useful to further elucidate the protein´s role in brain injury.
I have a few questions for authors.
Major point:
- Both in the MS title and in subtitles of Results and Discussion sections authors refer to long-term exposure to S100B, or increase of S100B. However, the correct description is that of Tg mice overexpressing S100B. I understand the group has a previous work reporting higher levels of the protein (circulating and in brain tissue), but they have not actually reported that in the present manuscript. Would you please elaborate on that?
Minor points:
- Abstract – line 12: please state you used male and female mice; - line 23: please consider “.. model in evaluating..”.
- Page 4, line 109 – please change “quantify” for “estimate”, since the method you used gives an estimation of cell numbers (as you explained in Methods).
- Page 6, line137 – 138 – please delete the phrase “ * indicates..”, since there are no differences in the variables depicted.
- Page 7, line 163 – does your model expresses 11 or 12 copies (in Abstract and Methods) of the gene?
- Page 9, lines 187 to 189, and 205 to 206 – in both sentences you say that presented results validate S100B long term treatment or examine long-term S100B exposure. However, you do not present data on S100B CSF or serum levels in the Tg mice.
Author Response
First of all, we would like to thank the Editors and Reviewers for all their time and input helping us to improve our manuscript.
Reviewer 1.
This is a very nice contribution of Dr L Rodrigues on transgenic mice overexpressing S100B protein. Assuming this is a model of long-term exposure to S100B, authors claim that S100tg mice have enhanced hippocampal progenitor cell proliferation, an effect that was not accompanied by reactive astrogliosis.
Experimental design is straightforward and methods used are adequate. Briefly, juvenile, adult and old-aged (1-y-old) Wt and Tg mice had cell proliferation estimated in the subgranular zone and the granular cell layer of the dentate gyrus, as assessed by BrdU and Ki67 markers. GFAP, ApoE, BDNF and RAGE were also studied, with enzymatic or western blot methods. There was an increase in progenitor cell proliferation in the subgranular zone in all age groups studied. Signaling was studied in 1-y-old mice; a decrease of ApoE content was found in the hippocampus and frontal cortex of Tg males, although its levels in serum, CSAF and in the adipose tissue did not differ. BDNF content decreased in the frontal cortex, but not in the hippocampus, of females, and GFAP hippocampus levels were decreased on both male and female Tg mice. Authors conclude that enhanced dentate neurogenesis in S100Tg mice is not accompanied by reactive astrogliosis and that sex dimorphism on the effects measured on ApoE and BDNF require further studies.
Results are original and shed light on S100B actions in the Tg model that might be useful to further elucidate the protein´s role in brain injury.
Does the introduction provide sufficient background and include all relevant references? Can be improved.
Reply 1: Thank you for very much for your positive comments. Based on your concern regarding background and relevant references, we added/revised the second paragraph of the Introduction.
I have a few questions for authors.
Major point:
- Both in the MS title and in subtitles of Results and Discussion sections authors refer to long-term exposure to S100B, or increase of S100B. However, the correct description is that of Tg mice overexpressing S100B. I understand the group has a previous work reporting higher levels of the protein (circulating and in brain tissue), but they have not actually reported that in the present manuscript. Would you please elaborate on that?
Reply2: We revised the title and elaborated on the interaction of S100B expression and "long-term increased S100B" levels in the second paragraph of the Introduction: “Recently, we reported increased S100B levels in brain tissue, cerebrospinal fluid (CSF) and serum of S100B transgenic animals, and pursue further research to understand the biochemical effects of the S100B overexpression. We refer to the term "long-term increased S100B" levels throughout the text with regard to conceivable (patho)physiological effects, keeping in mind that the transgenic animals overexpress theprotein and that the resulting S100B levels range in the nM order, far below toxic μM levels, used in cell culture.”
Minor points:
- Abstract – line 12: please state you used male and female mice;
- line 23: please consider “.. model in evaluating..”.
Reply 3: We made the change.
- Page 4, line 109 – please change “quantify” for “estimate”, since the method you used gives an estimation of cell numbers (as you explained in Methods).
Reply 4: We made the change.
- Page 6, line137 – 138 – please delete the phrase “ * indicates..”, since there are no differences in the variables depicted.
Reply 5: We made the change.
- Page 7, line 163 – does your model expresses 11 or 12 copies (in Abstract and Methods) of the gene?
Reply 6: We corrected the typo.
- Page 9, lines 187 to 189, and 205 to 206 – in both sentences you say that presented results validate S100B long term treatment or examine long-term S100B exposure. However, you do not present data on S100B CSF or serum levels in the Tg mice.
Reply 7: According to your suggestion, we eliminated the term “treatment” and are consistent with our wording as explained in the second paragraph of the Introduction. We changed to “we validated the effect long-term increased nanomolar S100B concentrations in S100Btg mice – as demonstrated in an earlier study – ” and “we examined whether the long-term increased S100B nanomolar levels promote”.
Reviewer 2 Report
The manuscript is potentially nteresting, but it is very descriptive. Further research needs to be done for it to be published on IJMS.
Author Response
First of all, we would like to thank the Editors and Reviewers for all their time and input helping us to improve our manuscript.
Reviewer 2.
Is the research design appropriate? Must be improved.
The manuscript is potentially interesting, but it is very descriptive. Further research needs to be done for it to be published on IJMS.
Reply: We thank the reviewer for evaluating our manuscript. We agree, that the manuscript is mainly descriptive. However, we aimed to investigate basic changes in this animal model allowing us to evaluate the specific contribution of increased S100B levels in acute brain injury in the future. We modified the second paragraph of the Introduction to clarify the aim as well as the relevance of the study:
”Recently, we reported increased S100B levels in brain tissue, cerebrospinal fluid (CSF) and serum of S100B transgenic animals [5], and pursue further research to understand the biochemical effects of the S100B overexpression. We refer to the term "long-term increased S100B" levels throughout the text with regard to conceivable (patho)physiological effects, keeping in mind that the transgenic animals overexpress the protein and that the resulting S100B levels range in the nM order, far below toxic μM levels used in cell culture [1]. In the present study, we continue earlier acute studies demonstrating nM S100B levels to enhance hippocampal neurogenesis and cognitive function following experimental brain injury and, even more importantly, in naïve animals [11,13-15].”
Further experimental studies in this transgenic animal model in situations of injury (e.g. TBI), will allow a detailed characterization of the role of S100B, involved signaling pathways (e.g. RAGE activation), and physiological effects.
Reviewer 3 Report
The authors provide evidence for S100B expression leads to increased increased progenitor cell proliferation in the SGZ and migration. They also noted interesting sex difference which merit further discussion in the text.
Minor corrections line 285 change "peomoted neurogenesis"
Line 290 consider change Vice versa to In contrast
Author Response
First of all, we would like to thank the Editors and Reviewers for all their time and input helping us to improve our manuscript.
Reviewer 3.
The authors provide evidence for S100B expression leads to increased increased progenitor cell proliferation in the SGZ and migration. They also noted interesting sex difference which merit further discussion in the text.
Minor corrections line 285 change "promoted neurogenesis"
Line 290 consider change Vice versa to In contrast
Reply: We cordially thank the reviewer for evaluating our manuscript and your suggestions. We changes the text accordingly.
Round 2
Reviewer 2 Report
The paper is descriptive. Other experiments will be performed.